# Deep proteome profiling reveals signatures of age and sex differences in paw skin and sciatic nerve of naïve mice

Feng Xian[†], Julia Regina Sondermann[†], David Gomez Varela, Manuela Schmidt*

Systems Biology of Pain, Division of Pharmacology & Toxicology, Department of Pharmaceutical Sciences, Faculty of Life Sciences, University of Vienna, Vienna, Austria

**Abstract** The age and sex of studied animals profoundly impact experimental outcomes in biomedical research. However, most preclinical studies in mice use a wide-spanning age range from 4 to 20 weeks and do not assess male and female mice in parallel. This raises concerns regarding reproducibility and neglects potentially relevant age and sex differences, which are largely unknown at the molecular level in naïve mice. Here, we employed an optimized quantitative proteomics workflow in order to deeply profile mouse paw skin and sciatic nerves (SCN) – two tissues implicated in nociception and pain as well as diseases linked to inflammation, injury, and demyelination. Remarkably, we uncovered significant differences when comparing male and female mice at adolescent (4 weeks) and adult (14 weeks) age. Our analysis deciphered protein subsets and networks that were correlated with the age and/or sex of mice. Notably, among these were proteins/biological pathways with known (patho)physiological relevance, e.g., homeostasis and epidermal signaling in skin, and, in SCN, multiple myelin proteins and regulators of neuronal development. Extensive comparisons with available databases revealed that various proteins associated with distinct skin diseases and pain exhibited significant abundance changes in dependence on age and/or sex. Taken together, our study uncovers hitherto unknown sex and age differences at the level of proteins and protein networks. Overall, we provide a unique proteome resource that facilitates mechanistic insights into somatosensory and skin biology, and integrates age and sex as biological variables – a prerequisite for successful preclinical studies in mouse disease models.

**\*For correspondence:**
manuela_schmidt@univie.ac.at

[†]These authors contributed equally to this work

**Competing interest:** The authors declare that no competing interests exist.

## Editor's evaluation

This study sheds light on the importance of appropriate experimental design for mouse disease models which has been overlooked so far. The authors provide solid evidence for dynamic changes of proteomes in mouse tissues according to age and sex. This type of work is extremely valuable to many biomedical scientists in the field for conducting reproducible research, especially in preclinical studies.

## Introduction

The age and sex of mice are major confounders in preclinical studies, affecting experimental outcomes across scales: from molecular, morphological, and physiological to behavioral parameters (*Flórez-Vargas et al., 2016*; *Flurkey et al., 2007*; *Fu et al., 2013*; *Jackson et al., 2017*). In mice, the first 12 weeks of life are characterized by pronounced changes in terms of growth and development of all organs and systems. Therefore, the Jackson Laboratory (https://www.jax.org) considers the widely used mouse strain C57BL/6J of mature adult physiology only at 12 weeks of age (*Flurkey et al.,*

*2007*). Similarly, the sex of mice needs to be considered when comparing experimental outcomes. Despite recently enforced policies by funding agencies to include animals of both sexes, most preclinical studies still do not perform experiments on male and female rodents in parallel, exhibit gaps in data analysis by sex, and often pool animals of both sexes and a wide range of ages (between 4 and 20 weeks) (*Flórez-Vargas et al., 2016*) given time and financial constraints (*Garcia-Sifuentes and Maney, 2021*; *Woitowich et al., 2020*). These practices may negatively impact reproducibility across studies, increase data variability, conceal differences or generate artifactual results, and, consequently, hamper translationally oriented preclinical research (*Flórez-Vargas et al., 2016*; *Jackson et al., 2017*; *Oliva et al., 2020*).

A prominent example of the enormous diversity of age ranges in publications are studies on rodent (mainly mice and rats) skin and peripheral sensory neurons (e.g., the sciatic nerve [SCN]) in the context of somatosensation and pain. Here, it is particularly noteworthy that often different age ranges were used for in vivo versus in vitro investigations. Mouse behavior experiments assessing paw sensitivity have routinely been performed in mice aged between 6 and 20 weeks (*Hanack et al., 2015*; *Moehring et al., 2018*; *Zheng et al., 2019*). Studies in cultured peripheral sensory neurons or keratinocytes have used mice aged 4–6 weeks (*Hanack et al., 2015*; *Poole et al., 2014*), 4–8 weeks (*Zheng et al., 2019*), 7–10 weeks (*Narayanan et al., 2018*; *Narayanan et al., 2016*), or 8–16 weeks (*Sadler et al., 2020*). Similarly, myelination of the SCN has been studied biochemically in mice aged 3 weeks (*Siems et al., 2020*), 10 weeks, and up to several months (depending on disease severity) (*Siems et al., 2021*). In contrast, cultured Schwann cells are generally derived from newborn rats (*Siems et al., 2020*). We have recently discovered a previously unknown age dependence of tactile sensitivity in the back skin and hind paws of mice (*Michel et al., 2020*). In particular, 4-week-old adolescent mice were more sensitive to innocuous tactile stimulation than 12-week-old adult mice. Interestingly, these observations correlated with similar changes in the activity of the mechanically activated ion channel Piezo2 and age-dependent transcriptome changes in peripheral sensory neurons.

Even so, to date, we still lack comprehensive knowledge about the differential molecular setup of the somatosensory system in dependence on age and sex, in particular on the level of the proteome. This is highly relevant as transcript levels only show limited correspondence with protein levels, which renders the functional interpretation of transcriptome results difficult, in particular under dynamic conditions such as development, maturation, and disease (*Liu et al., 2016*; *Schwanhäusser et al., 2011*; *Wang et al., 2017*). However, in contrast to well-established RNA-seq approaches, deep proteome profiling of complex tissues is still challenging, above all, for low abundant and transmembrane proteins. Latest technological advances in mass spectrometry (MS) and data analysis provide new solutions for these challenges (*Demichev et al., 2020*; *Meier et al., 2020*; *Meier et al., 2018*). Here, we thoroughly compared two MS-based quantitative proteomics approaches: commonly used data-dependent acquisition (DDA) paired with parallel accumulation serial fragmentation (DDA-PASEF) (*Meier et al., 2018*) compared to data-independent acquisition (DIA-PASEF) (*Meier et al., 2020*). The latter has been shown to offer superior performance for deep profiling (*Meier et al., 2020*), yet it has, thus far, only been applied by specialized laboratories given its high demands regarding technology and data analysis.

The goal of this work was to comprehensively catalog the protein setup of mouse paw skin and SCN, changes upon age (comparing adolescents, 4 weeks of age, and adults, 14 weeks of age), and differences between male and female wild-type (WT) C57BL/6J mice. The SCN is affected by a wide variety of motor and sensory neuropathologies induced by inflammation, trauma, and demyelination. Similarly, the skin, as our interface to the outer world, can be impaired by several inflammatory diseases like atopic dermatitis, psoriasis, and lupus erythematodes. In addition, both the skin and SCN are involved in nociception and (chronic) pain. We therefore focused on the potential implication of our data for preclinical research on skin- and SCN-related pathologies, including pain. Our results decipher hitherto unknown age and sex dependency of assorted proteins and signaling pathways, including those with known disease relevance. Taken together, our dataset is unique as (1) it provides a quantitative protein catalog of skin and SCN and (2) it does so in dependence on the age and sex of naïve mice. Given the heterogeneity of mouse age ranges in biomedical studies and the impact of age and sex on experimental outcomes, our results represent a highly valuable resource to foster future investigations in the context of skin and peripheral nerve (patho)physiology by enhancing reproducibility and unmasking hitherto unknown differences.

## Results

### DIA-PASEF allows deep and reproducible proteome profiling of mouse paw skin and sciatic nerves (SCN)

In this study, we analyzed 16 biological replicates of paw skin and SCN samples to compare the proteome between (1) two age groups, that is, 4-week-old adolescent mice and 14-week-old adult mice, and (2) males and females (*Figure 1—figure supplement 1*). To enable and optimize deep proteome profiling, we compared two label-free quantification strategies of MS-based quantitative proteomics. In particular, DDA-PASEF and DIA-PASEF. For each sample, we analyzed technical duplicates using a timsTOF Pro mass spectrometer (Bruker Daltonik). DDA or DDA-PASEF have been the methods of choice for most proteomics studies published so far (*Aballo et al., 2021*; *Aebersold and Mann, 2016*; *Meyer, 2021*). However, recent advances highlighted the superior performance of DIA-PASEF methods (*Brunner et al., 2022*; *Meier et al., 2021*), which we tested in our study side-by-side. Given the long acquisition time of approximately 20 days for all samples and replicates, we constantly monitored the performance of our MS setup in DIA-PASEF mode by using pooled skin peptides and SCN peptides as quality controls. Pearson's correlation coefficients were calculated for all quality control runs (*Figure 1A and B*). The average correlations of quality controls were 0.98 and 0.99 for pooled skin and SCN samples, respectively, indicating highly consistent stability of the instrument setup. Usually, DIA data is searched against a peptide library constructed from data obtained via DDA of the same sample; therefore, only proteins present in the library can be identified and quantified (*Ludwig et al., 2018*). In contrast, DIA-NN, a recently developed program based on deep neural networks, extensively advanced DIA workflows with a library-free database search mode (*Demichev et al., 2020*). Thus, we compared DDA-PASEF data subjected to a standard MaxQuant search (*Cox et al., 2014*) with DIA-PASEF data subjected to DIA-NN library-free search. As shown in *Figure 1C and D*, protein identifications from DDA-PASEF were highly covered by DIA-PASEF experiments, and DIA-PASEF detected additional 4135 and 3926 protein groups in skin and SCN, respectively (*Figure 1—source data 1 and 2*). Besides comparing protein identifications (protein IDs: for the remainder of this article, we will refer to protein groups as protein IDs for the sake of simplicity), we also compared both acquisition modes with respect to reproducibility at the quantitative level. Notably, we observed smaller coefficients of variation (CVs) across all DIA-PASEF runs (*Figure 1E and F*), indicating higher reproducibility compared to DDA-PASEF. Taken together, DIA-PASEF exhibited superior performance and was therefore chosen for further analysis of skin and SCN samples.

### Age-dependent protein abundance changes in mouse paw skin and SCN

In paw skin, we quantified > 8600 protein IDs across experimental groups (*Figure 2A*, *Figure 1—source data 1*). Comparing this proteome dataset with the most comprehensive (human) skin proteome dataset (*Dyring-Andersen et al., 2020*) published so far, our skin proteome covered approximately 70% (*Figure 2B*). Importantly, in our study we analyzed whole-skin lysates without preanalytical sample fractionation (e.g., separation of different skin layers) (*Dyring-Andersen et al., 2020*). Note that the previously published skin proteome was obtained from human hairy skin, while we analyzed mouse glabrous skin known to exhibit several differences in skin structure (*Gudjonsson et al., 2007*). Nonetheless, we identified all 50 known keratins and 19 collagens. In addition to structural proteins, we also quantified 13 members of the interleukin (IL) family and 11 of the S100 family (*Figure 2E*), known to play essential roles in the context of inflammation and infection (*Kozlyuk et al., 2019*; *Velazquez-Salinas et al., 2019*). Their detection across all skin samples with only a few missing values (note that we did not impute any data; see 'Materials and methods' for details) further validates the high performance and reproducibility of our optimized workflow. In SCN, approx. 8400 protein IDs were quantified across experimental groups (*Figure 2C*, *Figure 1—source data 2*). SCN harbor myelinated axons, which are closely associated with glia cells such as Schwann cells. Remarkably, the myelin proteome was nearly completely covered in our SCN data (94%); 1014/1077 described myelin proteins (*Siems et al., 2020*; *Figure 2D*), without a priori myelin enrichment as required in previous studies (*Siems et al., 2020*). Among the 63 proteins of the myelin proteome, which were not covered in our dataset, were ATP synthases, histones, and septins (*Figure 2—source data 1*). Another indication as to the depth and high performance of our workflow is the fact that we robustly quantified

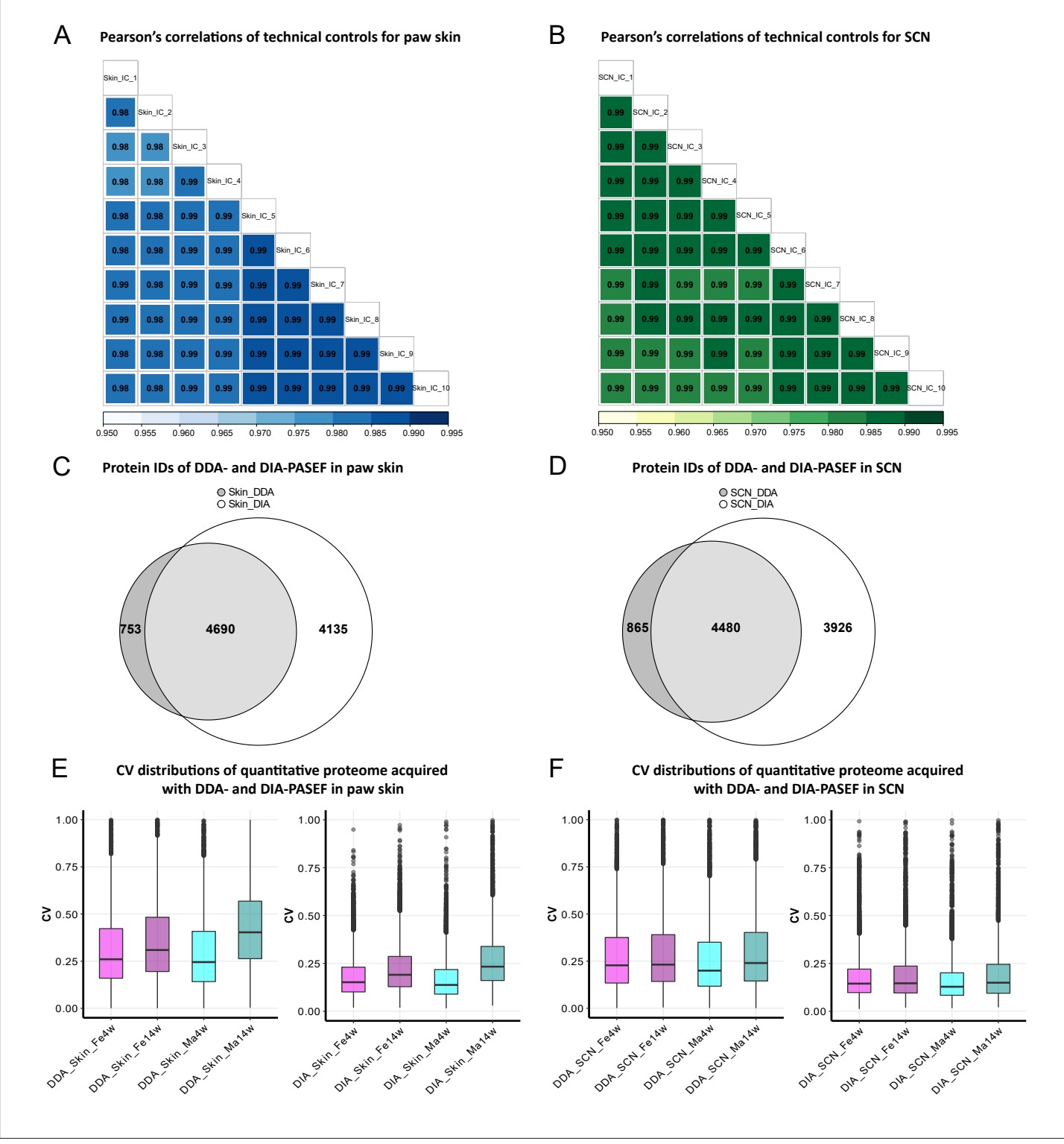

**Figure 1.** Data-independent acquisition paired with parallel accumulation serial fragmentation (DIA-PASEF) acquisition followed by DIA-NN analysis outperforms data-dependent acquisition paired with PASEF (DDA-PASEF) acquisition in deep proteome profiling of paw skin and sciatic nerve (SCN) of naïve mice. (**A, B**) Pearson's correlations of technical controls of paw skin (blue) and SCN (green) acquired over 20 days on a timsTOF Pro. (**C, D**) Comparisons of identified protein groups (protein IDs) using DDA- and DIA-PASEF workflows in paw skin (**C**) and SCN (**D**). (**E, F**) Coefficient of variation (CV) distributions of quantitative proteomes using DDA- and DIA-PASEF in paw skin (**E**) and SCN (**F**) of 4-week and 14-week-old males (cyan) and females (magenta).

*Figure 1 continued on next page*

*Figure 1 continued*

The online version of this article includes the following source data and figure supplement(s) for figure 1:

**Source data 1.** Quantitative proteome and differentially expressed protein (DEP) lists of paw skin.

**Source data 2.** Quantitative proteome and differentially expressed protein (DEP) lists of sciatic nerve (SCN).

**Figure supplement 1.** Experimental workflow of proteome profiling in mouse paw skin and sciatic nerves.

multiple ion channels across SCN samples (*Figure 2F*, *Figure 1—source data 2*), such as Trpv1 and several voltage-gated sodium channels (e.g., Scn8a, Scn9a, Scn11a) – again without requiring preanalytical membrane preparations. These ion channel identifications further corroborate the high quality of our approach as ion channels are usually expressed at low abundance and are notoriously difficult to be detected by MS given their pronounced hydrophobicity (*Samways, 2014*).

We employed principal component analysis (PCA) to visualize proteome similarities and differences across age and sex groups. Importantly, we only considered those proteins that were robustly quantified in all samples (according to all our quality criteria; see 'Materials and methods' for details), resulting in 6086 protein IDs in the skin and 6065 protein IDs in SCN (*Figure 2G and H*, *Figure 1—source data 1 and 2*). Age groups were clearly segregated by the first and second components in skin and SCN samples, indicating that age is a prominent discriminator in our study and associated differences can be tackled by whole-proteome analysis. Furthermore, to elucidate changes in abundance profiles across all experimental groups, fuzzy C-means clustering analysis was performed based on the average intensity of any protein ID quantified (*Figure 2—figure supplement 1*). Among the nine clusters generated, most of the proteins showed strong age patterns, such as clusters 2 and 5 in skin, and clusters 4, 6, and 7 in SCN. On the contrary, several proteins exhibited different expression trends in age/sex groups. For instance, most proteins in cluster 6 of the skin proteome showed minor age-dependent changes in females, while their abundance was notably increased in 14-week males compared to 4-week males (*Figure 2—figure supplement 1*). Similar sex-specific changes were also observed in SCN represented by cluster 3. Taken together, the clustering analysis of the paw skin and SCN proteome reveals thus far unknown expression patterns dependent on the biological variables age and sex, that is, sex-specific and -overlapping age dependency, which may affect mouse (patho) physiology.

## Diverse biological pathways exhibit age dependence intertwined with sex differences in paw skin

To explore this age dependency further, we applied a fold change (FC) cut-off (absolute log2 FC ≥ 0.585, i.e., an absolute FC of 1.5) in addition to a significance cut-off (q-value ≤ 0.05) and found 234 and 94 differentially expressed proteins (DEPs) in female and male skin datasets, respectively (*Figure 1—source data 1*). As shown in *Figure 3A*, 46 DEPs were shared between sexes, while 188 and 48 DEPs were unique for female and male skin (*Figure 1—source data 1*). Gene Ontology Biological Process (GO-BP) analysis of 46 common DEPs resulted in three significantly enriched pathways (criteria: at least four DEPs/pathway, Bonferroni-adjusted p-value ≤ 0.05). DEPs annotated to enriched pathways were mapped back to quantitative proteomic data, and the agglomerated z-scores of the pathways are visualized in *Figure 3B*, revealing a marked age-dependent pattern. As expected, skin development-related pathways such as 'protein hydroxylation' and 'collagen fibril organization' were enriched in 4-week skin compared to 14 weeks. These pathways were reported to be implicated in skin stability during development (*Rappu et al., 2019*). Specifically, several proline/serine hydroxylases (e.g., P4ha2, P3h1, P4ha1) were highly expressed in 4-week skin together with members of collagens (*Figure 3B*). Performing GO-BP enrichment on DEPs from age-dependent comparisons in female versus male mice (*Figure 3—figure supplement 1A and B*) revealed interesting biological insights into sex-dependent differences. In male skin, pathways of 'notch signaling' and 'extrinsic apoptotic signaling' were significantly enriched at 4 weeks, while 'actin-mediated cell contraction' and 'cellular component assembly involved in morphogenesis' were enriched at 14 weeks (*Figure 3C*). In female skin, proteins annotated to multiple interconnected pathways were significantly enriched at 14 weeks compared to 4 weeks (*Figure 3D*, *Figure 3—figure supplement 1A*). Many of these have been shown to contribute to skin homeostasis (*Hamanaka et al., 2013*; *Sreedhar et al., 2020*) such as 'plasma membrane organization', 'cotranslational protein targeting to membrane', and 'positive

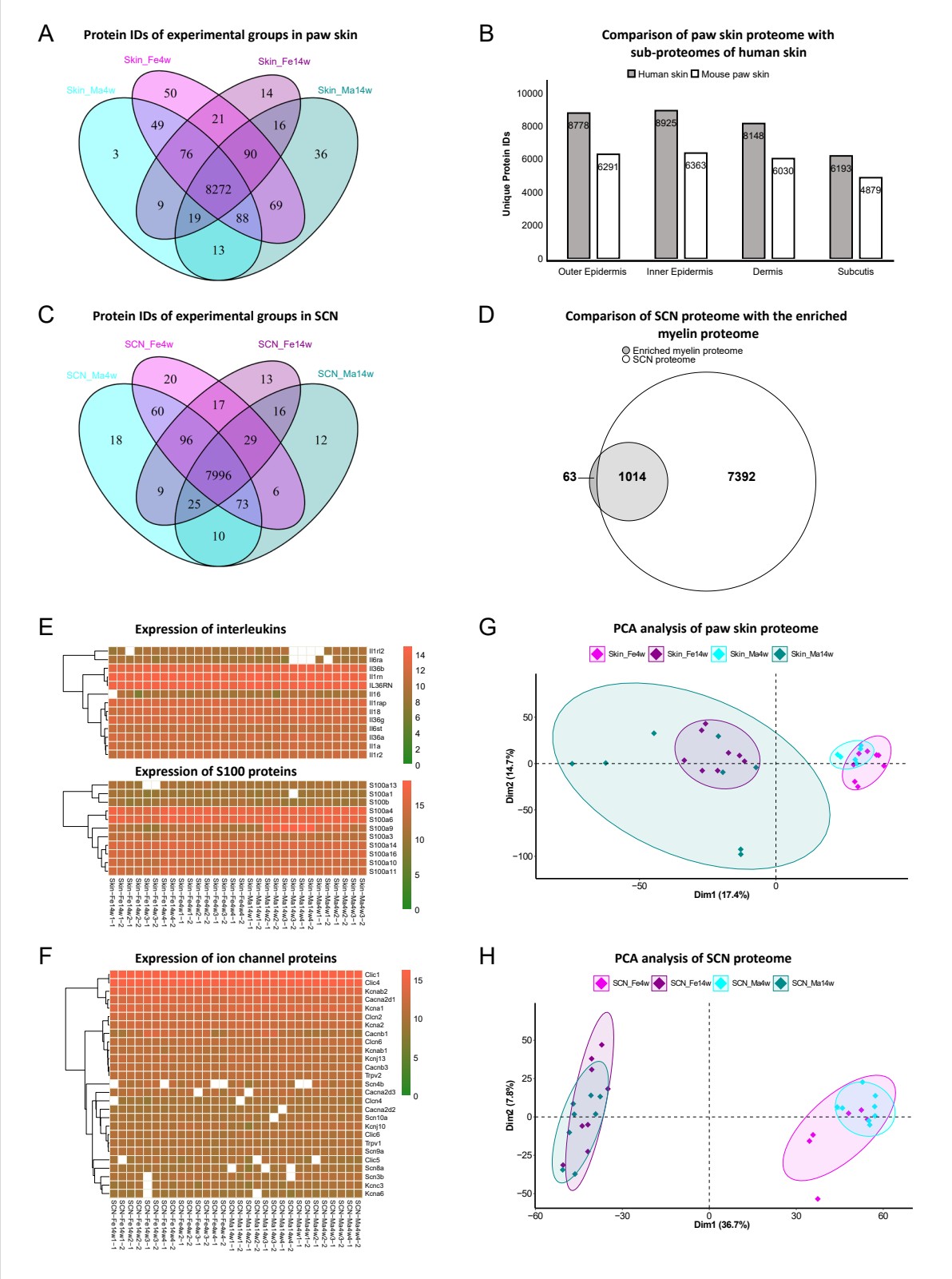

**Figure 2.** Age and sex differences in proteomes of paw skin and sciatic nerve (SCN). (**A**) Venn diagram shows unique and shared protein IDs across age and sex groups of paw skin. (**B**) Comparison of the quantified paw skin proteome with previously reported sub-proteomes of human skin (***Dyring-Andersen et al., 2020***) indicates high coverage in our proteome data. (**C**) Venn diagram shows unique and shared protein IDs across age and sex groups of SCN. (**D**) Our SCN proteome dataset harbors 1014 myelin proteins, i.e. 94% of the previously reported myelin proteome (***Siems et al., 2020***).

*Figure 2 continued on next page*

*Figure 2 continued*

(**E**) Heatmaps show the expression of interleukin and S100 protein families across all paw skin samples. (**F**) Heatmap shows the expression of ion channel proteins quantified across all SCN samples. Color legends are coded based on log2-transformed protein intensities. (**G, H**) Principal component analysis (PCA) reveals age as a prominent variable in paw skin and SCN tissues.

The online version of this article includes the following source data and figure supplement(s) for figure 2:

**Source data 1.** List of myelin proteins (*Siems et al., 2020*) not quantified in the sciatic nerve (SCN) proteome.

**Figure supplement 1.** Fuzzy C-means clustering analysis of the paw skin (**A**) and sciatic nerve (SCN) (**B**) proteome.

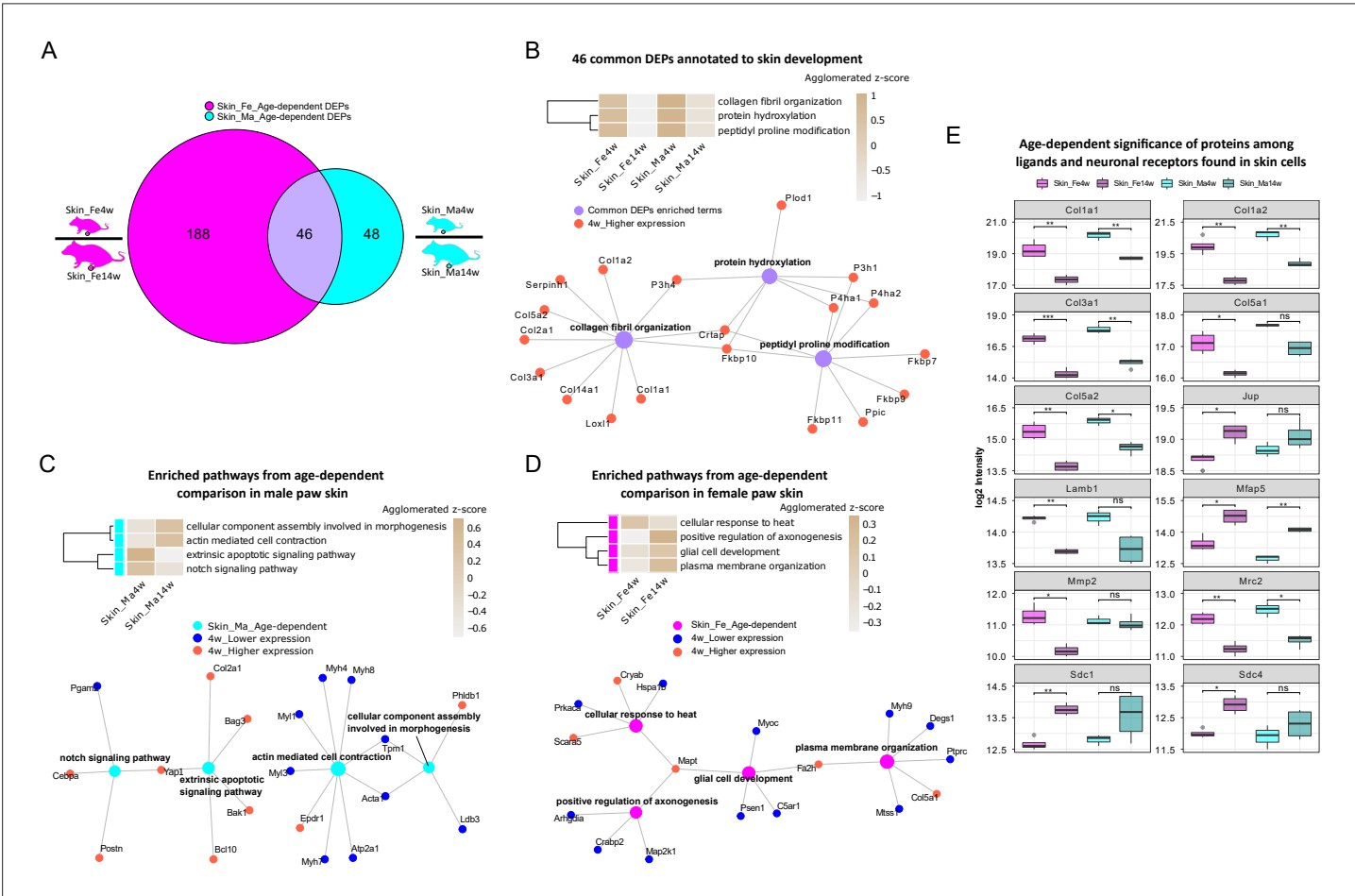

**Figure 3.** Differential expression analysis of paw skin samples reveals diverse age-dependent biological pathways in male and female mice. (**A**) Venn diagram illustrates unique and shared differentially expressed proteins (DEPs; criteria: q-value ≤ 0.05, absolute log2 fold change [FC] ≥ 0.585, i.e., an absolute FC of 1.5) from age-dependent comparisons in female (magenta) and male (cyan) paw skin. (**B**) 46 common DEPs (**A**) are annotated to pathways related to skin development. The agglomerated z-score of each pathway is visualized in the heatmap. Common DEPs are annotated to three interconnected pathways. All proteins annotated here were highly expressed in 4-week paw skin (red filled circle). (**C, D**) Enriched interconnected pathways from age-dependent comparison in male (cyan) and female (magenta) mice. Red: higher expression at 4 weeks; blue: lower expression at 4 weeks. (**E**) Ligands and neuronal receptors found in skin cells (*Wangzhou et al., 2021*) are significantly regulated by age. Significance levels are indicated as ns, q-value > 0.05, *q-value ≤ 0.05, **q-value ≤ 0.01, ***q-value ≤ 0.001, and ****q-value ≤ 0.0001.

The online version of this article includes the following source data and figure supplement(s) for figure 3:

**Source data 1.** List of ligands and neuronal receptors found in skin cell types (*Wangzhou et al., 2021*), which we quantified in the paw skin proteome.

**Figure supplement 1.** Gene Ontology Biological Process (GO-BP) analysis of differentially expressed proteins (DEPs) from age-dependent comparisons in female (**A**) and male (**B**) paw skin.

regulation of map kinase activity' besides others like 'positive regulation of axonogenesis' and 'glial cell development' (*Figure 3D*, *Figure 3—figure supplement 1A*). In contrast, proteins annotated to 'cellular response to heat' showed a higher z-score in 4-week skin of females.

Keratinocytes are among the most abundant cell types in skin, followed by fibroblasts, endothelial cells, melanocytes, and subsets of resident innate and adaptive immune cells. In addition, sparsely distributed sensory nerve endings in the skin play significant roles for aspects of somatosensation, including the detection of different physical stimuli, whether they be innocuous or noxious. However, this cellular diversity cannot be separated on the experimental level when analyzing complex tissue lysates as in our study. Therefore, we assessed the depth of our profiling workflow across different cell types indirectly by applying a recently published resource on ligand–receptor interactions in 42 cell types, including sensory neurons of mouse dorsal root ganglia (DRG) (*Wangzhou et al., 2021*). We extracted ligand–receptor interactions found across skin cell types (*Wangzhou et al., 2021*) for comparison with our skin dataset. In total, 144 ligands and receptors of DRG were present in our skin dataset (*Figure 3—source data 1*), of which 12 were significantly regulated (q-value ≤ 0.05) when comparing 4-week to 14-week mice (*Figure 3E*). For example, the ligand Lamb1 was found to be more abundant in 4-week female skin. Lamb1 was reported to serve as an anchor point for end feet of radial glial cells and as a physical barrier to migrating neurons (*Radmanesh et al., 2013*). Three receptors of Lamb1, low-density lipoprotein receptor-related protein 1 (Lrp1), C-type mannose receptor 2 (Mrc2), and suppressor of tumorigenicity 14 protein homolog (St14), were also identified in our dataset (*Figure 1—source data 1*). Interestingly, Mrc2 showed age-dependent statistical significance with higher expression at 4 weeks of age. Taken together, our results generally raise awareness of pronounced age dependency of protein expression in naïve mice, which should be carefully considered when pooling wide-ranging age groups in mouse studies.

## Prominent age and sex dependency of immune pathways and myelin proteins in SCN

In the SCN proteome, we observed similar age dependency as in paw skin. Differential expression analysis uncovered 929 DEPs and 1269 DEPs in age-dependent comparisons of female and male SCN (*Figure 4A*, *Figure 1—source data 2*), accounting for almost one-fifth of the here quantified SCN proteome. Pathway enrichment for 641 common DEPs (*Figure 1—source data 2*) and age-enriched DEPs in males versus females (for 4 weeks and 14 weeks, respectively) is given in *Figure 4—figure supplement 1*, spanning diverse categories from metabolic processes and translation to neuronal function and inflammatory/immune signaling. For example, among common DEPs, 'vesicle localization' had a higher z-score in 4-week SCN of both sexes, while pathways related to 'neuron survival', 'neurotransmitter transport', and 'anterograde axonal transport' were more pronounced in 14-week SCN of both sexes (*Figure 4B*). These processes appear to be interconnected via distinct DEPs (*Figure 4B*), suggesting crosstalk during development. For instance, superoxide dismutase (Sod1) was found to be less expressed in 4-week mice and represents a connecting hub of two pathways related to nervous system function (*Figure 4B*) in line with its implication in amyotrophic lateral sclerosis (ALS) (*Pansarasa et al., 2018*). Remarkably, 144 DEPs from age-dependent comparisons were associated with the 'synapse' as revealed by querying SynGO, a public reference for synapse research (*Koopmans et al., 2019*; *Figure 4—source data 1*).

Miscellaneous immune cell types are known to be present in the SCN, where they contribute to nerve health, damage, and repair, as well as to sensory phenomena of pain (*Kalinski et al., 2020*). Thus, we cross-referenced our data to the aforementioned ligand–receptor database (*Wangzhou et al., 2021*) and searched our SCN dataset for ligands of neuronal receptors known to be expressed in immune cells. Among the 56 immune cell ligands of neuronal receptors quantified in the SCN proteome, 19 showed age-dependent abundance changes in both sexes such as Agrin (Agrn), Thy-1 membrane glycoprotein (Thy1), and several collagens (*Figure 4C*, *Figure 4—source data 2*). Given their age-dependent abundance differences already in naïve mice, our results caution to adequately pool age groups when assessing immune signaling in mouse disease models as data might get skewed by underlying – and thus far unknown – age differences.

We also assessed ligands of neuron receptors *Wangzhou et al., 2021* found in glial cells and vice versa given their utmost importance for SCN (patho)physiology. We found 85 glial cell ligands of neuron receptors and 70 neuron ligands of glial cell receptors in our SCN proteome, and, more

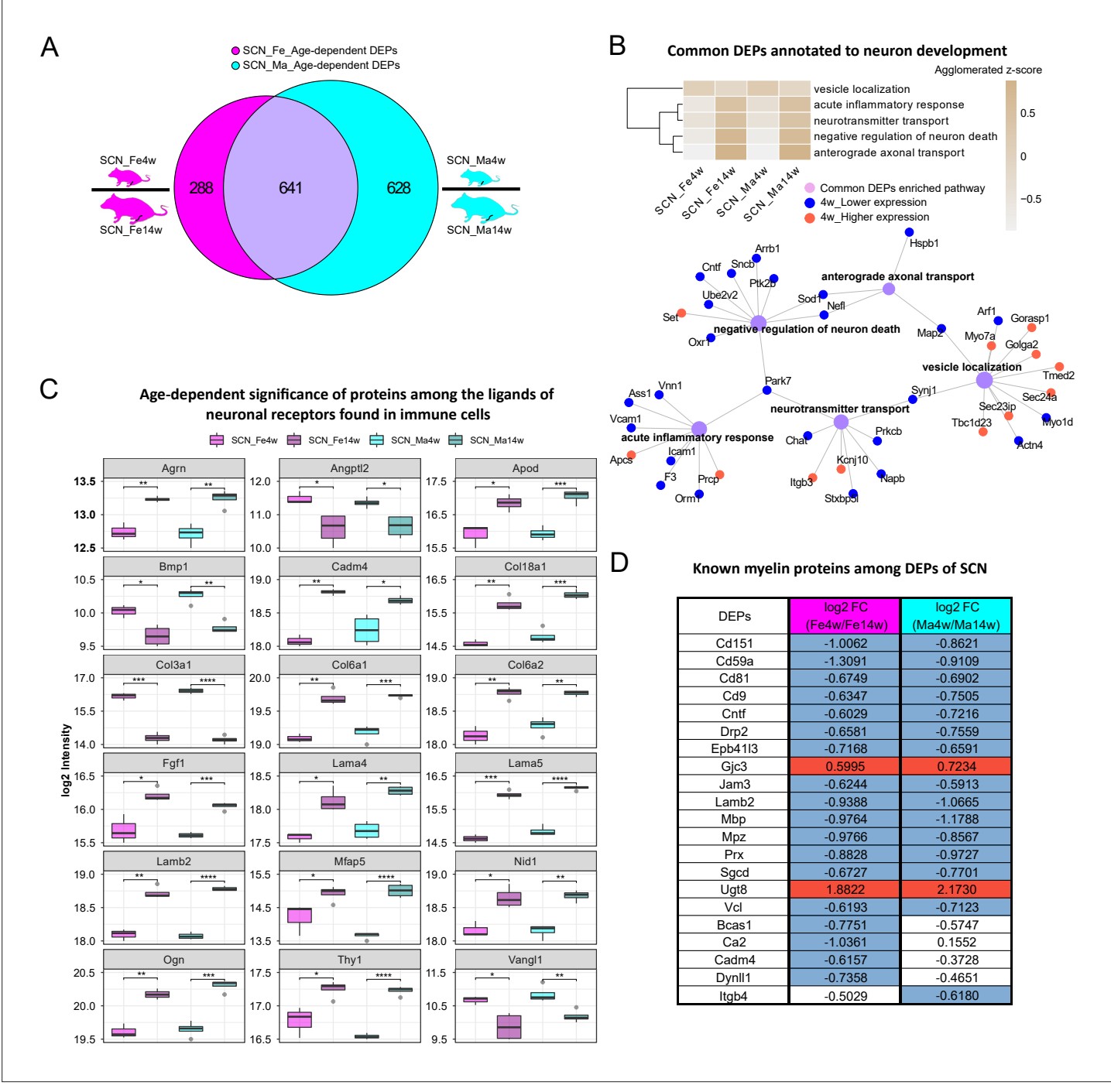

**Figure 4.** Age-dependent differential expression analysis in sciatic nerve (SCN) samples. (**A**) Venn diagram illustrates unique and shared differentially expressed proteins (DEPs) (criteria: q-value ≤ 0.05, absolute log2 fold change [FC] ≥ 0.585, i.e., an absolute FC of 1.5) from age-dependent comparisons in female (magenta) and male (cyan) SCN. (**B**) Common DEPs are annotated to pathways related to neuronal function and inflammation. The agglomerated z-score of each pathway is visualized in the heatmap. Red: proteins more abundant at 4 weeks; blue: proteins less expressed at 4 weeks. (**C**) Eighteen ligands of neuronal receptors found in immune cells (*Wangzhou et al., 2021*) are significantly regulated by age. Significance levels are indicated as: ns, q-value > 0.05, *q-value ≤ 0.05, **q-value ≤ 0.01, ***q-value ≤ 0.001, and ****q-value ≤ 0.0001. (**D**) Log2 FC of previously reported myelin proteins (*Siems et al., 2020*) in our age-dependent SCN datasets. Red: higher expression at 4 weeks; blue: lower expression at 4 weeks; white: not significantly regulated.

The online version of this article includes the following source data and figure supplement(s) for figure 4:

**Source data 1.** Synaptic proteins among differentially expressed proteins (DEPs) from age-dependent comparisons in sciatic nerve (SCN).

*Figure 4 continued on next page*

*Figure 4 continued*

**Source data 2.** Ligand list of neuronal receptors found in immune cells (*Wangzhou et al., 2021*), which we quantified in the sciatic nerve (SCN) proteome.

**Source data 3.** Neuronal ligands of glial receptors (*Wangzhou et al., 2021*), which we quantified in the sciatic nerve (SCN) proteome.

**Source data 4.** Glial ligands of neuronal receptors (*Wangzhou et al., 2021*), which we quantified in the sciatic nerve (SCN) proteome.

**Figure supplement 1.** Enriched Gene Ontology Biological Process (GO-BP) terms and their activity scores in age-dependent comparisons using common differentially expressed proteins (DEPs) (**A**) and DEPs enriched in male sciatic nerve (SCN) (**B**) and female SCN (**C**), respectively.

interestingly, about one-third of ligand-receptor pairs showed strong age dependency (*Figure 4—source data 3 and 4*), e.g., limbic system-associated membrane protein (Lsamp), a glial cell ligand mediating selective neuronal growth and axon targeting (*Sanz et al., 2017*), and two of its neuronal receptors, netrin-G1 (Ntng1) and thy-1 membrane glycoprotein (Thy1). While Lsamp exhibited higher expression in 4-week SCN than in 14-week SCN, its two receptors showed the opposite (*Figure 4—source data 4*, datasheet 2). Similar expression trends were also found in neuron ligands of glial cell receptors, for example, the ligand disintegrin and metalloproteinase domain-containing protein 23 (Adam23) were less abundant in 4-weeks SCN, but the receptor integrin beta-3 (Itgb3) was more abundant in 4-week SCN (*Figure 4—source data 3*, datasheet 2). These data may suggest a homeostatic mechanism specifically in young SCN to counterbalance ligand abundance on the receptor level – an intriguing hypothesis given that, for example, Lsamp has been reported to suppress neuronal outgrowth of DRG (*Sanz et al., 2017*).

In this respect, it is noteworthy that 21/90 known myelin proteins within the myelin proteome (*Siems et al., 2020*; *Figure 2D*) were differentially regulated by age, including highly abundant structural myelin proteins such as myelin basic protein (Mbp) and myelin protein P0 (Mpz) (*Figure 4D*). In line with previous reports in mice and zebrafish (*Siems et al., 2021*), Mbp and Mpz were significantly enriched in both male and female 14-week SCN, reflecting myelin assembly and axonal development with age. Similarly, CD59A glycoprotein (Cd59a), a sparsely expressed myelin protein associated with protection against complement-mediated lysis (*Zeis et al., 2016*), was more abundant at adult age (*Figure 4D*) as described previously in mouse brains (*Siems et al., 2021*). This congruency with published data on both high- and low-abundant myelin proteins validates our datasets and highlights their quality and depth of profiling. Note, however, that some myelin DEPs appear to be specific for female (Bcas1, Ca2, Cadm4, Dynll1) or male (Itgb4) SCN in dependence on age – a fact that has not been investigated in previous studies, which mostly focused on male mice (*Siems et al., 2021*). Further investigation of these changes correlated with sex and age will likely help to better understand the molecular setup of myelin and, importantly, associated pathologies.

## Sexual dimorphism in paw skin and SCN proteomes within distinct age groups

In addition to thus far presented sex differences in age-dependent proteome changes, we then turned to specifically looking at sex dependency within one age group in our datasets, that is, we compared skin and SCN proteomes between male and female mice at 4 weeks and 14 weeks, respectively. While several studies have addressed sexual dimorphism in animal models, most previous reports relied on investigating differences of transcript abundance in paw skin and nerve tissues (*Mecklenburg et al., 2020*; *Ray et al., 2019*). In our proteome dataset, we observed 58 DEPs (in skin) and 33 DEPs (in SCN) in a sex-dependent manner within the same age group (*Figure 5A and B*, *Figure 1—source data 1 and 2*). These numbers are low compared to aforementioned prominent changes upon age (*Figures 3A and 4A*). Of note, sex-dependent DEPs differed by age. For example, sex-dependent changes were much less pronounced in 4-week compared to 14-week skin (*Figure 5A*) and SCN (*Figure 5B*) in line with our initial PCA (*Figure 2G and H*). Interestingly, we did not observe any sex-dependent DEPs at 4 weeks in SCN (*Figure 5B*, orange circle). However, it is worth mentioning that all here reported DEPs are highly dependent on the experimental and analytical conditions of our study, for example, the chosen analysis and cut-off criteria, which are outlined in detail in 'materials and methods'. Nonetheless, PCA using only sex-dependent DEPs (*Figure 1—source data 1 and 2*) enabled effective discrimination between female and male samples (*Figure 5C and D*), suggesting that these DEPs might represent sex-specific protein signatures in mouse paw skin and SCN.

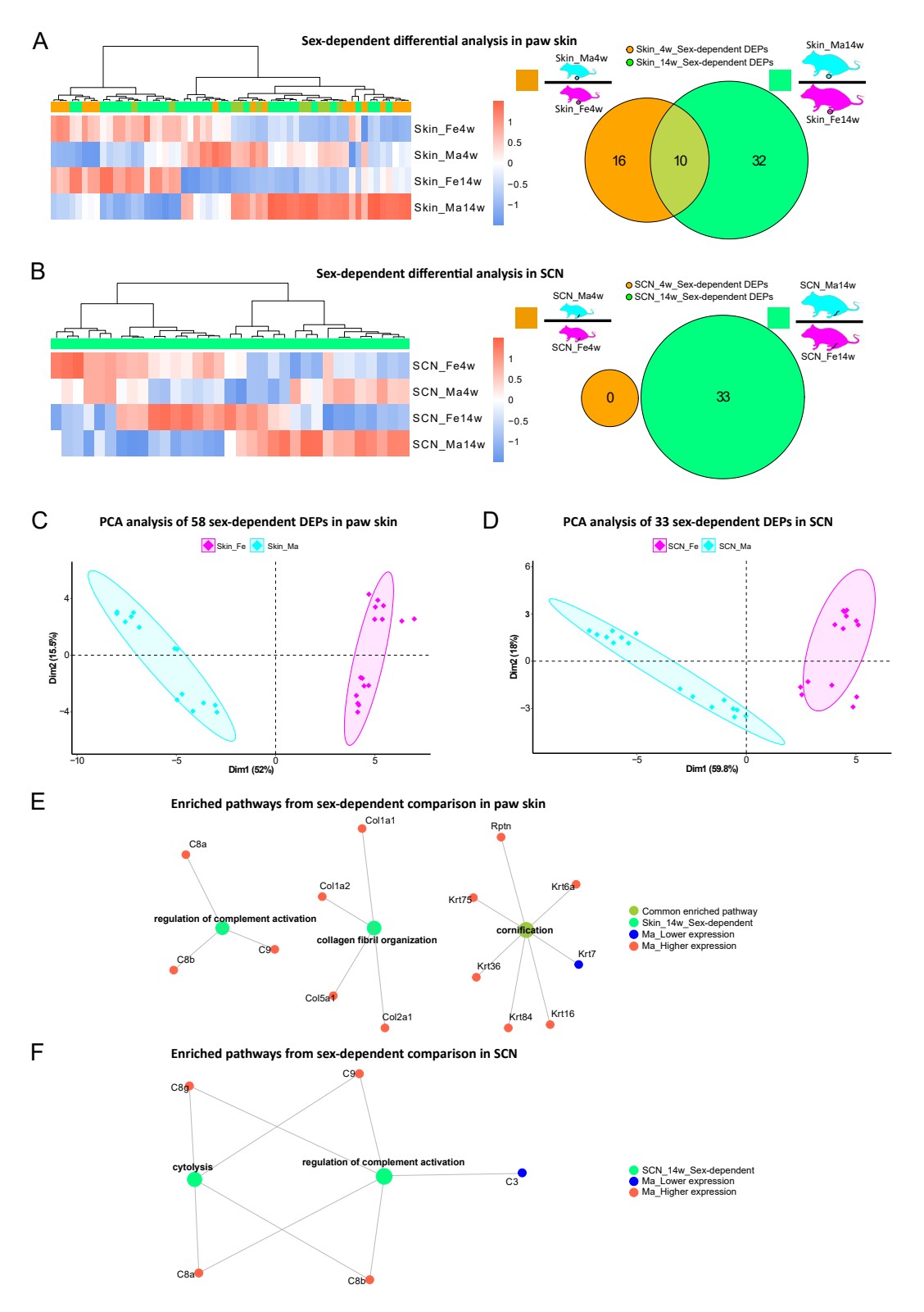

**Figure 5.** Differential expression analysis indicates protein signatures of sexual dimorphism in paw skin and sciatic nerve (SCN). (**A, B**) Differentially expressed proteins (DEPs) of sex-dependent comparisons at 4 weeks and 14 weeks in paw skin (**A**) and SCN (**B**). Heatmaps show the normalized protein expression (averaged intensity) across age and sex groups. Venn diagram depicts sex-dependent DEPs at 4 weeks (orange; note that none were found in SCN) and 14 weeks (green). (**C, D**) Principal component analysis (PCA) using DEPs of sex-dependent comparisons (in contrast to PCA on all identified

*Figure 5 continued on next page*

*Figure 5 continued*

proteins illustrated in *Figure 2G and H*) reveals sex as an effective discriminator in paw skin and SCN tissues; females (magenta) and males (cyan). (**E, F**) Visualization of enriched pathways using sex-dependent DEPs at 4 weeks and 14 weeks. Red: higher expression in males; blue: lower expression in males; green: pathways enriched at 14 weeks in a sex-dependent manner.

The online version of this article includes the following source data and figure supplement(s) for figure 5:

**Source data 1.** Sex-associated transcripts of human tibial nerves (*Ray et al., 2019*), which we quantified in the sciatic nerve (SCN) proteome.

**Source data 2.** Sex-associated transcripts of mouse hind paws (*Mecklenburg et al., 2020*), which we quantified in the paw skin proteome.

**Figure supplement 1.** Sexual dimorphism in SCN and paw skin.

Overall, GO-BP analysis revealed that sex-dependent DEPs in skin and SCN could be annotated to few (owing to the low number of DEPs) but distinct pathways (*Figure 5E and F*). For example, 'regulation of complement activation' was significantly enriched in both skin and SCN in sex comparisons at 14 weeks (*Figure 5E and F*). The complement system is known to be crucially implicated in immunity, host defense, inflammation, and associated pathologies (*Merle et al., 2015*; *Ray et al., 2019*). Thus, revealing its sexual dimorphism provides a crucial guide for adequate experimental design in future studies, especially when using rodent disease models.

Next, we compared our datasets with previously published literature on molecular sex differences. Unfortunately, we did not find any resource that assessed sex differences in mouse SCN. Therefore, we compared our SCN datasets with sex-associated genes found in human tibial nerve (*Ray et al., 2019*; *Figure 5—source data 1*). Among these 149 differentially expressed genes (DEGs), we quantified 31 proteins in our SCN dataset, of which only 20 were differentially regulated. However, these proteins rather exhibited age- and not sex-dependent differences in our mouse SCN data, potentially due to species differences (*Ray et al., 2019*; *Figure 5—figure supplement 1A*). A previous transcriptomics study on mouse hind paw skin revealed 123 DEGs comparing female with male mice aged 8–12 weeks (*Mecklenburg et al., 2020*). Among these 123 DEGs, we quantified 42 proteins in our skin dataset, of which 14 were sexually dimorphic at 4 weeks and/or 14 weeks (*Figure 5—source data 2*; *Figure 5—figure supplement 1B*) in line with published data (*Mecklenburg et al., 2020*). Reasons as to why we did not observe sex-dependent differences in the remaining 28 out of these 42 proteins could be manifold: starting with the broad age range used in the transcriptome study (*Mecklenburg et al., 2020*) to the known fact that transcript levels only show limited correspondence with protein expression levels (*Liu et al., 2016*; *Maier et al., 2009*; *Reimegård et al., 2021*). The latter highlights the importance of performing profiling studies on the proteome and integrate these data with other -omics approaches.

## Multiple proteins associated with skin diseases and pain exhibit age and sex dependency

In light of translational research, reverse translation of human data to mouse models of skin diseases is of high utility. Therefore, we compared our skin datasets with a list of top candidates (skin disease transcriptomic profiles, https://biohub.skinsciencefoundation.org/) found to be regulated in the skin of human patients suffering from skin diseases such as psoriasis, acne, atopic dermatitis, and rosacea. Intriguingly, our proteome results harbor 329 out of 907 disease genes, of which 27 proteins showed significant age and/or sex dependency across varied skin diseases (*Figure 6A*, *Figure 6—source data 1*). Fuzzy C-means clustering analysis of these 329 skin disease-related proteins revealed not only discrete abundance patterns among age and sex groups but also proteins with differential profiles when comparing all four experimental groups (*Figure 6—source data 2*). For example, proteins in cluster 6 showed higher abundance in 14-week male skin, whereas proteins in cluster 7 exhibited lower abundance in 4-week skin of both sexes compared to 14 weeks (*Figure 6B*). Discrete abundance patterns in dependence on age and sex were also observed on the level of individual proteins (examples are given in *Figure 6C*; full list detailed in *Figure 6—source data 1*). For example, collagen alpha-1(V) chain (Col5a1), collagen alpha-2(V) chain (Col5a2), and coiled-coil domain-containing protein 80 (Ccdc80) were less abundant at 14 weeks in both male and female samples and, in parallel, showed distinct sex differences. In contrast, indolethylamine N-methyltransferase (Inmt) and keratin type II cytoskeletal 6A (Krt6a) represent examples with higher levels in both sexes at 14 weeks. Others display prominent sex dependency such as transcription factor Sp1 (Sp1) and versican core protein

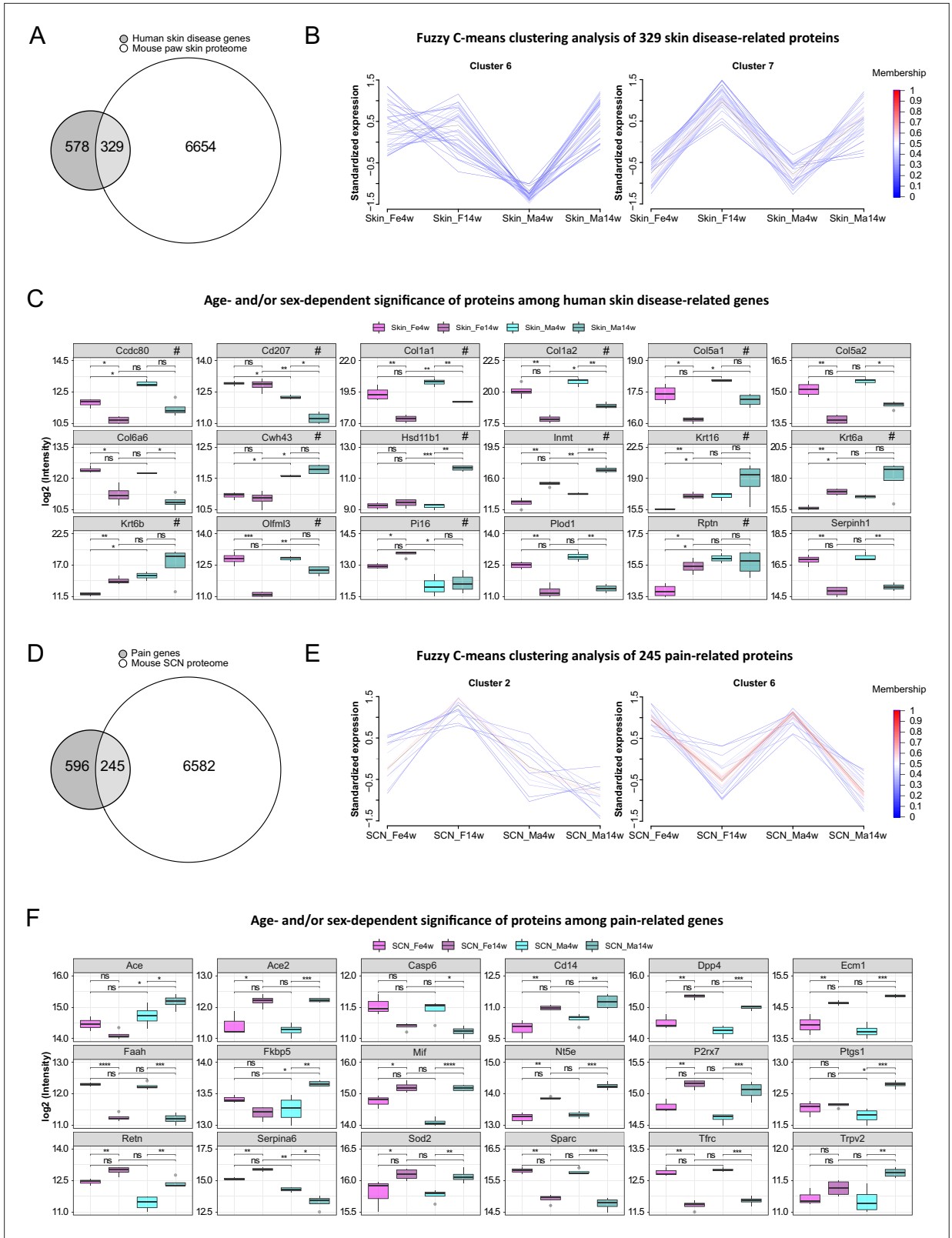

**Figure 6.** Multiple proteins associated with skin diseases and pain exhibit age and sex dependence. (**A**) Venn diagram indicates the number of quantified protein IDs in paw skin (white) associated with various human skin diseases (light gray) upon comparison with a skin disease database (dark gray, https://biohub.skinsciencefoundation.org/). (**B**) Examples of fuzzy C-means clustering analysis of the 329 protein IDs associated with human skin diseases illustrate their relative expression in experimental groups (other clusters are detailed in *Figure 6—source data 2*). (**C**) Significantly

*Figure 6 continued on next page*

*Figure 6 continued*

expressed protein IDs associated with human skin diseases show age and/or sex dependency. Proteins marked with '#' represent examples related to hand–foot psoriasis, palmoplantar pustulosis, and vesicular hand eczema. (**D**) Venn diagram indicates the number of quantified protein IDs in sciatic nerve (SCN) (white) associated with pain (light gray) upon comparison with known pain genes (dark gray). Pain-related genes were downloaded from publicly available pain gene databases: https://www.painresearchforum.org/, https://humanpaingeneticsdb.ca/hpgdb/, and http://paingeneticslab.ca/4105/06_02_pain_genetics_database.asp. (**E**) Examples of fuzzy C-means clustering analysis of the 245 protein IDs associated with pain illustrate their relative expression in experimental groups (other clusters are detailed in *Figure 6—source data 2*). (**F**) Significantly expressed protein IDs associated with pain show age and/or sex dependency. Significance levels in (**C**) and (**F**) are indicated as ns, q-value > 0.05, *q-value ≤ 0.05, **q-value ≤ 0.01, ***q-value ≤ 0.001, and ****q-value ≤ 0.0001.

The online version of this article includes the following source data for figure 6:

**Source data 1.** Gene candidates of various human skin diseases (https://biohub.skinsciencefoundation.org/), which we quantified in the paw skin proteome.

**Source data 2.** Fuzzy C-means clustering membership of 329 and 245 pathology-related proteins quantified in the paw skin and sciatic nerve (SCN) proteome.

**Source data 3.** Differentially expressed genes (DEGs) of hand–foot psoriasis, palmoplantar pustulosis, and vesicular hand eczema quantified in the paw skin proteome.

**Source data 4.** Pain-related genes (https://www.painresearchforum.org/, http://paingeneticslab.ca/4105/06_02_pain_genetics_database.asp, and https://humanpaingeneticsdb.ca/hpgdb/), which we quantified in the sciatic nerve (SCN) proteome.

**Source data 5.** Differentially expressed proteins (DEPs) upon sciatic nerve (SCN) injury (spared nerve injury [SNI] model of neuropathic pain in mice) (*Barry et al., 2018*) quantified in the SCN proteome.

(Vcan) being specifically regulated in female skin in an age-dependent manner (less abundant in 4-week female skin compared to 14 weeks; *Figure 6—source data 1*). Among skin diseases used here for comparison (*Figure 6—source data 1*), several are known to exhibit an autoimmune component such as alopecia areata, lichen plantus, lupus erythematosus, psoriasis, and vitiligo. Overall, autoimmune skin diseases are more prevalent in females. Thus, we specifically checked whether top gene signatures of aforementioned autoimmune-associated skin diseases are among the reported female-enriched proteome changes of our study (*Figure 1—source data 1*). Indeed, 11 proteins could be identified, of which 9 were differentially regulated by age only in females (*Figure 6—source data 1*, data sheet 3). For instance, transforming growth factor-beta-induced protein ig-h3 (Tgfbi) and transcription factor Sp1 (Sp1) were more abundant in 14-week female skin, but no significance was found in males. It is noteworthy that in our study we investigated hairless glabrous skin in mice. However, transcriptomic profiles of human skin diseases used for comparison are mostly derived from human hairy skin – a difference that needs to be taken into account when interpreting the presented comparisons between mouse and human skin. This is why we additionally selected transcriptomic studies on skin diseases affecting human glabrous skin, that is, palms and soles, such as handfoot psoriasis (*Ahn et al., 2018*), palmoplantar pustulosis (*McCluskey et al., 2022*), and vesicular hand eczema (*Voorberg et al., 2021*). Among top gene signatures presented in glabrous skin datasets (2498 DEGs), 928 proteins were quantified in our paw skin datasets and 96 of them showed age and/or sex dependency including several collagens (*Figure 6—source data 3*; examples are marked with '#' in *Figure 6C*). Overall, our data suggest significant regulation of human skin disease profiles by age and sex in mice – knowledge of utmost significance for reverse translational studies on the skin.

The SCN is affected by various pathologies such as nerve injury and neuropathic pain (*Chen et al., 2021*; *Hildreth et al., 2009*). In this context, we assessed the presence of known pain-related genes in SCN datasets by comparison with three publicly available pain gene databases (https://www.pain-researchforum.org/, http://paingeneticslab.ca/4105/06_02_pain_genetics_database.asp, and https://humanpaingeneticsdb.ca/hpgdb/). Among 841 pain genes, 245 were quantified in our SCN proteome (*Figure 6D*, *Figure 6—source data 4*). Similar to skin disease-related proteins, fuzzy C-means clustering analysis revealed distinct abundance profiles for these 245 pain-related proteins differing by age and/or sex (*Figure 6E*, *Figure 6—source data 2*). Intriguingly, 132 of these pain-related proteins displayed significant changes (q-value ≤ 0.05) by age and/or sex (*Figure 6—source data 4*), of which examples are given in *Figure 6F*. For instance, angiotensin I-converting enzyme (Ace) exhibits pronounced age and sex differences while its family member Ace2 only showed age differences (i.e., higher expression in adult mice). Ace2 has been described to be associated with increased risk of nonspecific orofacial symptoms in the OPPERA (*Smith et al., 2013*) prospective study, and Ace is

linked to migraine and potentially higher frequency as well as susceptibility (*Dandona et al., 2007*). Generally, Ace plays an essential role in vascular physiology and inflammation within the renin–angiotensin system. The FK506 binding protein 51 (Fkbp5) also displayed significant age and sex differences, with its expression being lower in female adults but higher in male adults compared to 4-week mice. Fkbp5 is a glucocorticoid receptor co-chaperone, and its polymorphisms predict persistent musculoskeletal pain after traumatic stress exposure (*Bortsov et al., 2013*; *Linnstaedt et al., 2018*). Moreover, it is generally involved in the acute stress response linked to stress-related disorders in humans via the hypothalamic-pituitaryadrenal (HPA) axis (*Häusl et al., 2021*). Interestingly, variations in serpin peptidase inhibitor clade A member 6 (Serpina6), another pain gene with pronounced age differences and sex-opposite expression, are also associated with the HPA stress axis impacting the susceptibility to musculoskeletal pain (*Holliday et al., 2010*), the risk of cardiovascular disease, as well as gene expression in peripheral tissues (*Crawford et al., 2021*). These few examples highlight the importance of considering changes in the abundance of aforementioned proteins across ages and in both sexes for research on preclinical disease models. Therefore, our datasets represent a highly valuable and unique resource for biomedical studies. To further illustrate this utility, we have critically inspected our previous SCN datasets derived from the neuropathic pain model of nerve injury (spared nerve injury [SNI] model) in male adult mice (*Barry et al., 2018*). Indeed, several candidate proteins we previously reported (*Barry et al., 2018*) to be regulated upon SNI in adult males, for example, angiotensin-converting enzyme (Ace), apolipoprotein E (Apoe), complement C3 (C3), progranulin (Grn), epidermal growth factor receptor (Egfr), and voltage-dependent calcium channel subunit alpha-2/delta-1 (Cacna2d1), exhibited age-dependent abundance differences in males but not in females (*Figure 6—source data 5*) – a fact that has not been known before and might crucially affect SNI-induced pathology. Taken together, our data emphasize the essential need for male versus female as well as age-matched biomedical studies in parallel.

## Discussion

Age and sex as parameters in rodent-based research are known to strongly affect experimental outcomes in vivo and in vitro. Despite this knowledge, the molecular setup of most cell and tissue types and how they are impacted by age and sex has not yet been discerned (*Garcia-Sifuentes and Maney, 2021*; *Woitowich et al., 2020*). This gap is especially eminent at the proteome level – a fact that hampers our understanding of the molecular signature underlying physiological processes and disease phenotypes. Here, we present an optimized workflow of quantitative proteomics, which utilizes DIA-PASEF followed by data analysis with the publicly available program DIA-NN (*Demichev et al., 2020*). This approach enabled us to deeply profile and quantify the proteome of paw skin and SCN in adolescent versus adult male and female mice. Strikingly, our data reveal unprecedented insights into hitherto unknown age and sex differences of biological processes relevant for skin and SCN physiology as well as associated disorders. Therefore, our work serves as unique resource for the scientific community by defining the protein compendium of mouse skin and SCN and changes thereof in dependence on age and sex. In addition, our results assertively highlight the significance of appropriate age and sex matching and provide a stepping stone for optimizing preclinical and translational research toward enhanced reproducibility and success.

We focused our analysis on paw skin and SCN isolated from 4-week and 14-week male and female mice. Both tissue types are crucially implicated in multiple diseases. On one hand, in various allergic, itchy, and inflammatory skin pathologies such as atopic dermatitis, psoriasis, and lupus erythematodes. The SCN, on the other hand, is affected by a wide variety of motor and sensory neuropathologies induced by inflammation, trauma, and demyelination. Moreover, both the skin and SCN are involved in nociception and pain, including chronic conditions. Our reasoning for the chosen age groups was as follows: given time and budget constraints as well as 'comparability to historical data' (*Reiber et al., 2022*), it has become standard practice to perform experimental studies with mice of a wide age range, generally from 4 weeks to 12 weeks of age (*Jackson et al., 2017*; *Reiber et al., 2022*), regardless of the studied biological system. Of note, mice aged 3–4 weeks are mostly used for cell culture-based in vitro studies (*Boyer et al., 1994*; *Isensee et al., 2017*; *Lin and Chen, 2018*; *Malin et al., 2007*). For example, peripheral sensory neurons of DRG exhibit better health and growth factor-dependent survival when isolated from young rodents (*Malin et al., 2007*; *Melli and Höke, 2009*). Similarly, Schwann cells originated from younger human donors proliferated faster than those

from older donors (*Boyer et al., 1994*; *Monje, 2020*). In contrast to these younger ages used for cell culture-based research, most studies on mouse behavior are conducted at ages of 6–12 weeks, that is, a period with overt maturational changes before reaching adulthood from 12 weeks of age onward (*Flurkey et al., 2007*). Notable examples of this practice include studies on cutaneous touch, somatosensation and (chronic) pain, and diseases of the central nervous system (CNS) and peripheral nervous system (PNS), just to name a few (*Narayanan et al., 2016*; *Poole et al., 2014*; *Zheng et al., 2019*). Furthermore, the most extensive and highly valuable RNA-seq-based resource describing the molecular setup of mouse CNS and PNS cell types (*Zeisel et al., 2018*) (https://www.mousebrain.org) was assembled by pooling mice of both sexes aged 2–3 weeks. Our results show that the proteome exhibits clear differences when comparing 4-week with 14-week mice raising concerns about the aforementioned practices of comparing and correlating in vitro data derived from young ages with in vivo data of older ages. Failed correlations among these age groups may have led to false negatives and prevented new findings. On the contrary, mechanisms discovered in young cells may be wrongly accounted for phenotypic differences observed in adult rodents. Based on our data, we strongly suggest suitable age matching across different methods within a study to ensure scientific rigor and reproducibility.

From a technical point of view, we have employed a highly sensitive workflow based on DIA-PASEF followed by data analysis via DIA-NN (*Demichev et al., 2020*). This enabled us to provide the most in-depth and highly reproducible proteome dataset of mouse skin and SCN published thus far. While we have previously used DIA-MS, but not DIA-PASEF, to investigate various tissues in mouse pain models (*Barry et al., 2018*; *Rouwette et al., 2016*; *Sondermann et al., 2019*), we have not achieved the high quantitative depth reported here. For instance, in SCN, we quantified more than twice as many proteins (~8400 IDs, *Figure 2C*) as in our previous study using DIA-MS (3141 IDs) (*Barry et al., 2018*). Even so, we would like to stress the fact that all here presented data and DEPs are closely interconnected with the experimental and analytical setup of our study. For example, datasets would become more comprehensive, if improved mass spectrometers, more biological replicates, lower cut-offs (in our study: absolute log2FC ≥ 0.585), or more powerful analysis algorithms (see discussion below) will be used in future investigations.

DIA workflows generally facilitate high reproducibility of quantitative profiling (*Domon and Aebersold, 2010*); however, resulting MS spectra are very complex requiring careful interpretation (*Bilbao et al., 2015*). To address this, improvements of diverse aspects have been continuously implemented. For instance, from the hardware point of view, the addition of ion mobility to chromatographic and mass separation of peptides has significantly reduced spectral complexity in DIA-MS (*Helm et al., 2014*). Furthermore, the DIA-PASEF workflow was developed on timsTOF Pro instruments (Bruker Daltonik) and enables nearly complete sampling of the precursor ion beam (*Meier et al., 2020*). Likewise, new algorithms have been developed to interpret complex spectra. For example, Arnaud Droit and colleagues systematically evaluated and compared DIA data processing software (*Gotti et al., 2021*) such as DIA-NN, DIA-Umpire, OpenSWATH, ScaffoldDIA, Skyline, and Spectronaut: DIA-NN outperformed others in terms of peptide and protein identifications. Given that DIA-PASEF has not yet been widely used, above all not for mouse skin and SCN, we have systematically compared the more commonly used DDA-PASEF mode with DIA-PASEF. We specifically compared (1) the number of protein IDs quantified, (2) the coefficients of variation (CVs), and (3) quantitative correlations between technical replicates of each sample. As expected, DIA-PASEF outperformed DDA-PASEF in all parameters (*Figure 1*). Consequently, we obtained all reported datasets on paw skin and SCN via DIA-PASEF followed by DIA-NN library-free data analysis.

Both analyzed tissues harbor various cell types such as keratinocytes, immune cells, peripheral nerve endings, and fibroblasts in skin, and glia cells, immune cells, and axons in SCN. Because of this cellular complexity, we cannot assign the detected age- and sex-dependent proteome differences to specific cell types – a limitation applicable to all -omics assays, if not performed on the single-cell level. In contrast to established procedures for single-cell RNA-seq, proteomics on single cells is still in its infancy. However, the transcriptome cannot predict disease phenotypes in a straightforward manner given the highly dynamic regulation of protein levels by manifold mechanisms (*Liu et al., 2016*; *Schwanhäusser et al., 2011*). Consequently, monitoring pathologies and associated changes in molecular and cellular signaling requires the integration of proteomics into a multi-omic suite. This calls for the development of technological advances for accurate, highly sensitive, and comprehensive

proteome profiling on the single-cell level. Recently, Brunner et al. established an ultra-high-sensitivity MS workflow to quantify single-cell proteomes from cultured HeLa cells with protein IDs ranging from 1018 (cell cycle G1 phase) to 1932 (G1/S phase) per single cell (**Brunner et al., 2022**). Though this pipeline has provided the opportunity to analyze single-cell-derived proteomes, the workflow requires elaborate sample preparation and technical equipment. Furthermore, it has not yet been applied to complex tissues harboring multiple cell types.

In conclusion, we present the most extensive proteome compendium of mouse skin and SCN described thus far. Our work demonstrates prominent and previously unknown sexual and age dimorphism in paw skin and SCN of naïve mice. Many of the reported differences are likely relevant for our mechanistic understanding of various disorders involving the skin (e.g., inflammatory pathologies like atopic dermatitis and psoriasis) and SCN (motor- and sensory neuropathologies induced by inflammation, trauma, and demyelination). Therefore, our study serves as a unique resource for different life science disciplines. In addition, our work advocates for the importance of appropriate age and sex matching and provides new avenues for improving the reproducibility, generalizability, and success of preclinical and translational research on skin and SCN.

# Materials and methods

## Key resources table

| Reagent type (species) or resource | Designation | Source or reference | Identifiers | Additional information |
|---|---|---|---|---|
| Strain, strain background (mouse) | C57BL/6J | In-house bred | | Wild type, female and male, 3-4 and 1415 weeks old |
| Chemical compound, drug | Acetonitrile | Fisher Scientific | 10001334 | |
| Chemical compound, drug | Formic acid | Fisher Scientific | 15658430 | |
| Chemical compound, drug | 10× PBS | Fisher Scientific | 11594516 | |
| Chemical compound, drug | Tris 1 M | Accugene/Avantor | 733-1653 | |
| Chemical compound, drug | Glycerol | Fisher Scientific | 10021083 | |
| Chemical compound, drug | Dithiothreitol 1 M | Sigma-Aldrich | 43816 | |
| Chemical compound, drug | Acetone | Sigma-Aldrich | 1000201000 | |
| Chemical compound, drug | Ethanol | Sigma-Aldrich | 1117272500 | |
| Chemical compound, drug | Iodoacetamide | Sigma-Aldrich | I1149 | |
| Chemical compound, drug | Ammonium bicarbonate | Sigma-Aldrich | 09830-500G | |
| Chemical compound, drug | Water MS grade | Sigma-Aldrich | 1.15333.1000 | |
| Chemical compound, drug | Trypsin/Lys-C | Promega | V5073 | |
| Chemical compound, drug | Trypsin | Serva | 37283.01 | |
| Chemical compound, drug | Sera-Mag SpeedBead beads | Cytiva | 65152105050250, 45152105050250 | 1:1 mix |
| Other | cOmplete Protease Inhibitor Cocktail | Roche/Merck | 58929700001 | Mix of protease inhibitors |
| Other | Protein LoBind tube | Eppendorf | 0030108116 | Reagent tube |
| Other | Aurora Series UHPLC column | IonOpticks | AUR2-25075C18A-CSI | 25 cm × 75 µm column |
| Other | Biopsy punch 4 mm | Kai Medical | 48401 | Skin biopsy punch |
| Software, algorithm | MaxQuant | Max Planck Institute of Biochemistry | | Version 1.6.17.0 |
| Software, algorithm | DIA-NN | https://github.com/vdemichev/DiaNN | RRID:SCR_022865 | Version 1.8.0 |
| Software, algorithm | R | https://www.r-project.org/ | | Version 4.1.1 |

*Continued on next page*

*Continued*

| Reagent type (species) or resource | Designation | Source or reference | Identifiers | Additional information |
|---|---|---|---|---|
| Software, algorithm | Mouse proteome database | UniProt | UP000000589 | Downloaded on 2021-07-08, 17070 entries |

## Reagents

All reagents were purchased from Sigma-Aldrich (St. Louis, MO) if not mentioned otherwise. Acetonitrile (ACN) and formic acid (FA) were purchased from Fisher Scientific (Hampton, New Hampshire; both FA and ACN were liquid chromatography-mass spectrometry [LC-MS] grade). LC-MS grade water from Sigma was used for all solutions.

## Animals and tissue isolation

In-house bred C57BL/6J mice of both sexes were used. Housing and sacrificing of mice were carried out with approval of the Max Planck Institute for Multidisciplinary Sciences Institutional Animal Care and Use Committee (IACUC, see 'Ethics statement'). All mice used in this study were group-housed in individually ventilated cages in a 12 hr light/dark cycle in the animal facility of the Max Planck Institute for Multidisciplinary Sciences with water and food ad libitum. Mice were sacrificed at ages 3-4 (referred to as 4-week-old mice) or 14-15 weeks (referred to as 14-week-old mice). Thus, the experiment consisted of four different conditions with four biological replicates each (4-week females, 4-week males, 14-week females, and 14-week males). After $CO_2$ euthanization of mice, the SCN and paw skin were isolated. SCN was rinsed in ice-cold PBS before flash-freezing in liquid nitrogen. For the paw skin, a 4 mm punch biopsy (Kai Medical, Solingen, Germany) of the plantar aspect of the paw was taken, and the dermis and epidermis were separated from underlying tendons/muscle tissue under a microscope. The flash-frozen tissue was stored at 80°C until further use. For both skin and SCN, tissue from two mice of the same sex and age were pooled together as one biological replicate.

## Protein extraction

For protein extraction, each SCN was cut into three pieces with a scalpel on a glass slide and transferred to a protein LoBind tube (Eppendorf, Hamburg, Germany) prefilled with 250 µL lysis buffer (100 mM Tris–HCl, 5% glycerol, 10 mM DTT, 2% SDS) and in presence of 1× cOmplete Protease Inhibitor Cocktail (Roche, Basel, Switzerland). Samples were then sonicated using Bioruptor Pico (Diagenode, Seraing, Belgium) for 15 cycles (30 s on and 30 s off, 4°C) at low frequency. After a short vortex, samples were further incubated at 70°C for 10 min with 1000 rpm agitation. Remaining tissue debris was removed after centrifugation at 10,000 × *g* for 5 min, and the supernatant was taken into a new tube. To remove lipids in the tissue lysates, 1250 µL (5× sample volume) of cold acetone was added, and the sample was placed at 20°C for 4 hr. With centrifugation at 14,000 × *g* for 30 min, acetone was removed, and proteins were collected at the bottom. The protein pellet was further washed with 1.5 mL cold ethanol (80% v/v) followed by 30 min centrifugation at 14,000 × *g*. The protein pellet was air-dried for 20 min at room temperature before the addition of 100 µL lysis buffer. A further incubation at 70°C for 10 min with 1000 rpm agitation was performed to solubilize all proteins. Protein concentrations were measured using NanoPhotometer N60 (Implen, Munich, Germany) at 280 nm, and 50 µg protein of each sample was taken for protein reduction (5 mM dithiothreitol [DTT], 30 min incubation at 60°C) and alkylation (20 mM iodoacetamide [IAA], 30 min at room temperature in the dark). The remaining IAA in the sample was quenched with addition of 5 mM DTT. Skin biopsies were cut into two pieces and homogenized in 350 µL lysis buffer with the help of a glass dounce. The homogenate was further solubilized by incubation at 70°C for 10 min with 1500 rpm agitation and sonication with the Bioruptor Pico (15 cycles, 30 s on and 30 s off, 4°C, low frequency). Removal of cell debris and subsequent steps were done as described for the SCN.

## SP3-assisted protein digestion and peptide clean-up

For protein clean-up and digestion, a modified version of the single-pot, solid-phase-enhanced sample preparation (SP3) method from Hughes et al. was used (*Hughes et al., 2019*). Briefly, 10 µL of pre-mixed Sera-Mag SpeedBead beads (Cytiva, Marlborough, MA) were added into 50 µg protein sample. To initiate binding of proteins to the beads, one volume of absolute ethanol was added

immediately, followed by incubation on a Thermomixer (Eppendorf) at 24°C for 5 min with 1000 rpm agitation. The supernatant was removed after 2 min resting on a magnetic rack, and the beads were rinsed three times with 500 µL of 80% ethanol. Rinsed beads were reconstituted in 50 µL digestion buffer (50 mM ammonium bicarbonate, pH 8). Protein digestion was performed with 2 µg of either sequencing grade trypsin (SCN samples) or trypsin/Lys-C (skin samples) for 18 hr at 37°C with 950 rpm agitation. After digestion, ACN was added to each sample to a final concentration of 95%. Mixtures were incubated for 8 min at room temperature and then placed on a magnetic rack for 2 min. The supernatant was discarded, and the beads were rinsed with 900 µL of 100% ACN. The rinsed beads were reconstituted either in 40 µL (SCN samples) or 20 µL (skin samples) LC-MS grade water to elute the peptides. Peptide concentration was measured in duplicate using NanoPhotometer N60 (Implen, München, Deutschland) at 205 nm. Peptide samples were acidified with FA to a final concentration of 0.1% and stored at -20°C until LC-MS/MS analysis.

## LC-MS/MS

Nanoflow reversed-phase liquid chromatography (Nano-RPLC) was performed on a NanoElute system (Bruker Daltonik, Bremen, Germany). Then, 250 ng of peptides were separated with a 130 min gradient on a 25 cm × 75 µm column packed with 1.6 µm C18 particles (IonOptics, Fitzroy, Australia). Mobile solvent A consisted of 2% ACN, 98% water, 0.1% FA, and mobile phase B of 100%, 0.1% FA. The flow rate was set to 400 nL/min for the first 2 min and the last 9 min of the gradient, while the rest of the gradient was set to 250 nL/min. The mobile phase B was linearly increased from 0 to 20% from 3 min to 110 min, flowed by a linear increase to 35% within 10 min and a steep increase to 85% in 0.5 min. Then, a flow rate of 400 nL/min at 85% was maintained for 9 min to elute all hydrophobic peptides. NanoElute LC was coupled with a hybrid TIMS quadrupole TOF mass spectrometer (timsTOF Pro, Bruker Daltonik) via a CaptiveSpray ion source. Each sample was analyzed in both DIA and DDA modes coupled with parallel accumulation serial fragmentation (PASEF) one after another in duplicate. The TIMS analyzer was operated in a 100% duty cycle with equal accumulation and ramp times of 100 ms each. Specifically, in DDA-PASEF mode (*Meier et al., 2018*), 10 PASEF scans were set per acquisition cycle with ion mobility range (1 /k0) from 0.6 to 1.6, and singly charged precursors were excluded. Dynamic exclusion was applied to precursors that reached a target intensity of 17,500 for 0.4 min. Ions with m/z between 100 and 1700 were recorded in the mass spectrum. In DIA-PASEF mode, precursors with m/z between 400 and 1200 were defined in 16 scans containing 32 ion mobility steps with an isolation window of 26 Th in each step with 1 Da overlapping for neighboring windows. The acquisition time of each DIA-PASEF scan was set to 100 ms, which led to a total cycle time of around 1.8 s (*Meier et al., 2020*). In both DDA and DIA-PASEF modes, the collision energy was ramped linearly from 59 eV at 1/k0 = 1.6–20 eV at 1/k0 = 0.6.

## DDA-PASEF data processing

All DDA data were analyzed with MaxQuant (version 1.6.17.0) and searched with Andromeda against *Mus musculus* database from UniProt containing 17,070 protein entries (downloaded on 2021-07-08). The minimal peptide length was set to six amino acids, and a maximum of three missed cleavages were allowed. The search included variable modifications of methionine oxidation and N-terminal acetylation, deamidation (N and Q), and fixed modification of carbamidomethyl on cysteine, and a maximum of three modifications per peptide were allowed. The 'Match between run' function was checked within 0.5 min retention time window and 0.05 ion mobility window. Mass tolerance for peptide precursor and fragments were set as 10 ppm and 20 ppm, respectively. The FDR was set to 0.01 at precursor level and protein level. Label-free quantification algorithm was used to quantify identified proteins with a minimum of one razor and unique peptide. The rest of the parameters were kept as default. Proteus, an R package (https://github.com/bartongroup/Proteus), was used for downstream analysis of MaxQuant output (*Gierlinski et al., 2018*).

## DIA-PASEF data processing

DIA-NN (*Demichev et al., 2020*) was used to process DIA-PASEF data in library-free mode with the same *M. musculus* proteome database to generate the predicted spectrum library. Trypsin/P was used for in silico digestion with an allowance of maximum three missed cleavages. A deep learning-based method was used to predict theoretical peptide spectra along with its retention time and

ion mobility. Variable modifications on peptides were set to N-term methionine excision, methionine oxidation, and N-term acetylation, while carbamidomethylation on cysteine was a fixed modification. The maximum number of variable modifications on a peptide was set to 3. Peptide length for the search ranged from 5 to 52 amino acids. Aligned with the DIA-PASEF acquisition method, m/z ranges were specified as 400–1200 for precursors and 100–1700 for fragment ions. Both MS1 and MS2 mass accuracy were set to 10 ppm as recommended. Unique genes were used as protein inference in grouping. RT-dependent cross-run normalization and Robust LC (high accuracy) options were selected for quantification. The main report from the DIA-NN search was further processed with the R package, DiaNN (https://github.com/vdemichev/diann-rpackage; *Demichev et al., 2020*; *Demichev, 2020*) to extract the MaxLFQ (*Cox et al., 2014*) quantitative intensity of gene groups for all identified protein groups with q-value < 0.01 as criteria at precursor and gene group levels.

## Visualization of proteomic data

Pearson's correlation plots were created with corrplot package (https://github.com/taiyun/corrplot; *Wei and Simko, 2022*). Venn diagrams were plotted using VennDiagram (https://CRAN.R-project.org/package=VennDiagram; *Chen, 2022*), and area-proportional Venn diagrams were created with eulerr package (https://github.com/jolars/eulerr; *Larsson, 2022*). All box plots and bar plots used to visualize the proteomic data were created using ggplot2 package (https://github.com/tidyverse/ggplot2; *Wickham et al., 2022*). In addition, the package ggpubr (https://rpkgs.datanovia.com/ggpubr/; *Kassambara, 2022*) was used for significance tests in comparisons within box plots. PCA was performed with the factoextra package (https://rpkgs.datanovia.com/factoextra/index.html; *Kassambara, 2020*), and proteins that were quantified in all biological and technical replicates are considered for the analysis. Color-coded tables were prepared in Microsoft Excel 2019.

## Differential expression analysis and protein function enrichment

Two samples (Skin_Ma4w-4 and SCN_Fe4w-1) were excluded from the following analysis due to significantly lower protein content than corresponding biological replicates. Only proteins quantified in ≥ 75% of replicates in each experimental group were submitted to differential expression analysis, resulting in 6983 protein IDs and 6827 protein IDs for paw skin and SCN proteomes, respectively. Technical duplicates of each biological sample were averaged. These quantitative data were imported into the R package, ProTIGY (https://github.com/broadinstitute/protigy; *Krug et al., 2021*), and log2-transformed, followed by normalization based on log2-mean intensity. Two-sample moderated *t*-test was used to test for statistical significance test of individual contrasts. Age-dependent comparisons (Fe = female; Ma = male): Skin_Fe4w versus Skin_Fe14w, Skin_Ma4w versus Skin_Ma14w, SCN_Fe4w versus SCN_Fe14w, and SCN_Ma4w versus SCN_Ma14w. Sex-dependent comparisons: Skin_Ma4w versus Skin_Fe4w, Skin_Ma14w versus Skin_Fe14w, SCN_Ma4w versus SCN_Fe4w, and SCN_Ma4w versus SCN_Fe4w. Proteins with the adjusted (Benjamini and Hochberg, for multiple testing) p-value ≤ 0.05 (hereafter referred to as the q-value) and the absolute log2 FC ≥ 0.585, that is, an absolute FC of 1.5, were considered as DEPs in each contrast. GO-BP enrichment and visualization of DEPs was performed using the pathfindR package (version 1.6.4, https://github.com/egeulgen/pathfindR; *Ulgen, 2022*) in R environment (*Ulgen et al., 2019*). In general, a threshold of ≥ 4 DEPs and adjusted p-value ≤ 0.05 (Bonferroni) was applied for significantly enriched pathways. Due to the low number of sex-dependent DEPs found in paw skin and SCN within one age group, a threshold of ≥ 3 DEPs and adjusted p-value ≤ 0.05 (Bonferroni) was applied for significantly enriched pathways in dependence of sex.

## Acknowledgements

We thank Tanja Nilsson (Max Planck Institute for Multidisciplinary Sciences, Göttingen, Germany) and Sabrina Grundtner (Division of Pharmacology & Toxicology, University of Vienna, Austria) for their efforts regarding tissue isolation. We are grateful to the Systems Biology of Pain team at the University of Vienna for discussions and Daniel Segelcke (University Hospital Muenster, Muenster, Germany) for

critically reading the manuscript. We also appreciate the help of the Bruker Daltonik team for setting up the proteomics platform.

## Additional information

### Funding

| Funder | Grant reference number | Author |
|---|---|---|
| Universität Wien | | Julia Regina Sondermann |
| Max Planck Society | | Manuela Schmidt |

The funders had no role in study design, data collection and interpretation, or the decision to submit the work for publication.

### Author contributions

Feng Xian, Data curation, Formal analysis, Validation, Investigation, Visualization, Methodology, Writing – original draft, Writing – review and editing; Julia Regina Sondermann, Data curation, Formal analysis, Validation, Investigation, Methodology, Writing – original draft, Writing – review and editing; David Gomez Varela, Conceptualization, Validation, Project administration, Writing – review and editing; Manuela Schmidt, Conceptualization, Data curation, Supervision, Funding acquisition, Validation, Writing – original draft, Project administration, Writing – review and editing

### Author ORCIDs

Feng Xian http://orcid.org/0000-0002-8345-0108
Julia Regina Sondermann http://orcid.org/0000-0002-6891-2159
David Gomez Varela http://orcid.org/0000-0003-2502-9419
Manuela Schmidt http://orcid.org/0000-0003-1972-3519

### Ethics

All mouse work strictly followed the regulations stated in the German animal welfare law (TierSchG §4). For this study, mice were sacrificed by $CO_2$ to obtain tissues, and no other procedure was performed. Therefore, mouse work of this study is not considered an animal experiment according to §7 Abs. 2 Satz 3 TierSchG. All procedures were approved and supervised by the animal welfare officer and the animal welfare committee of the Max Planck Institute for Multidisciplinary Sciences, Göttingen, Germany. The animal facility at the Max Planck Institute for Multidisciplinary Sciences is registered according to §11 Abs. 1 TierSchG.

### Decision letter and Author response

Decision letter https://doi.org/10.7554/eLife.81431.sa1
Author response https://doi.org/10.7554/eLife.81431.sa2

## Additional files

### Supplementary files

• MDAR checklist

### Data availability

All datasets included in this study are listed in 14 source data files. Moreover, proteome raw data generated in this study were deposited to the PRIDE archive via ProteomeXchange (https://www.proteomexchange.org) with identifier PXD034476.

The following dataset was generated:

| Author(s) | Year | Dataset title | Dataset URL | Database and Identifier |
|---|---|---|---|---|
| Xian F, Sondermann JR, Gomez Varela D, Schmidt M | 2022 | Deep Proteome profiling reveals signatures of age and sex differences in paw skin and sciatic nerve of naïve mice | http://proteomecentral.proteomexchange.org/cgi/GetDataset?ID=PXD034476 | ProteomeXchange, PXD034476 |

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
