## [Editor Report]

This study sheds light on the importance of appropriate experimental design for mouse disease models which has been overlooked so far. The authors provide solid evidence for dynamic changes of proteomes in mouse tissues according to age and sex. This type of work is extremely valuable to many biomedical scientists in the field for conducting reproducible research, especially in preclinical studies.

---

## [Decision Letter]

**Decision letter after peer review:**

Thank you for submitting your article "Deep Proteome Profiling Reveals Signatures of Age and Sex Differences in Paw Skin and Sciatic Nerve of Naïve Mice" for consideration by *eLife*. Your article has been reviewed by 3 peer reviewers, one of whom is a member of our Board of Reviewing Editors, and the evaluation has been overseen by Mone Zaidi as the Senior Editor. The following individual involved in review of your submission has agreed to reveal their identity: Aicha Asma Houfani (Reviewer #3).

Essential revisions:

This manuscript is interesting that sheds light on the importance of appropriate experimental design for mouse disease model which have been overlooked so far. This paper provided dynamic changes of proteomes in mouse tissues according to the age and sex which is interesting and worth publishing. The results look quite solid based on the proper methodology. This type of work is extremely valuable to many biomedical scientists in the field for conducting reproducible research, especially in preclinical studies.

The study was designed carefully however, the results for an association with skin diseases should need careful consideration since paw skins are usually not involved in those dermatoses.

1. Batch effect identification and correction using ComBat, SVA, RUV, harmony, and FastMNN etc. are extensively performed in (sc)RNA-seq datasets. Considering the amount of transcriptomics/proteomics datasets that were already produced without careful age/sex matching, the authors could have explored the possibility of age/sex (batch) correction in silico.

2. The authors failed to provide biochemical validation of DEPs identified from the work. I would appreciate that functional testing of the relevant changes to the mouse phenotypes, but this might be beyond the scope of this paper but do wonder whether relying on changes in protein expression level are truly enough to conclude the impact in the preclinical settings.

3. The impact of sex seems to be not impressive based on the PCA analysis of proteome, especially compared to age (Figure 2G and H) and PCA plots using sex-dependent DEPs are misleading (Figure 5C and D).

4 Although the authors choose paw skin and associated SCN to interpret nociception and pain, in the skin research fields, paw skins are not usually investigated compared to ear and back skins, as there is a highly limited type of dermatoses involving palms and soles in human. In this regard, the current investigation of paw skins may result in the intrinsic limitation for interpretation of the results in translating human dermatoses including psoriasis, atopic dermatitis, and rosacea, which usually spare palms and soles. It would be more interesting and translationally reasonable if the authors select available gene signatures for other dermatoses typically involving palms and soles, such as palmoplantar pustulosis or vesicular hand eczema.

5. Although the results for age-dependent changes of proteome profiles in paw skins are somewhat predictable (4w: immature skin, 14w: mature skin), sex-dependent difference in paw skin proteome is quite interesting. As female sex is more prone to autoimmune skin diseases such as lupus erythematosus, it would be more interesting if the authors further analyze whether sex-dependent proteome difference is enriched in autoimmune-associated gene signature.

---

## [Author Response]

Essential revisions:This manuscript is interesting that sheds light on the importance of appropriate experimental design for mouse disease model which have been overlooked so far. This paper provided dynamic changes of proteomes in mouse tissues according to the age and sex which is interesting and worth publishing. The results look quite solid based on the proper methodology. This type of work is extremely valuable to many biomedical scientists in the field for conducting reproducible research, especially in preclinical studies.The study was designed carefully however, the results for an association with skin diseases should need careful consideration since paw skins are usually not involved in those dermatoses.1. Batch effect identification and correction using ComBat, SVA, RUV, harmony, and FastMNN etc. are extensively performed in (sc)RNA-seq datasets. Considering the amount of transcriptomics/proteomics datasets that were already produced without careful age/sex matching, the authors could have explored the possibility of age/sex (batch) correction in silico.

We highly appreciate the reviewers´ valuable suggestion using batch effect correction as an alternative for related studies. Indeed, this suggestion offered an interesting quantitative measurement of the advantages and limitations of these correction technologies (see Author response image 1). We tested this exciting suggestion on our own datasets given the lack of accessible granular data from published proteomics studies.

We focused on the skin proteome of our study and divided it into sub-group1 (Skin_Ma4w pooled with Skin_Fe4w) and sub-group2 (Skin_Ma14w pooled with Skin_Fe14w), i.e. we considered the two sexes as different batches. In this setting, we aimed at investigating age-dependent proteome changes. The two sub-groups were corrected individually using ComBat (non-parametric adjustment) within the SVA package in R to try to remove batch effects. The corrected proteome data was further subjected to differential expression analysis in the ProTiGY package as described in the manuscript, and we generated new DEP lists from age-dependent comparisons.

As shown in Author response image 1 and B, the ComBat correction (grey circles) led to partly different DEPs compared with the original, i.e. age- and sex-matched data in our study (white circles).

Similarly, we investigated sex-dependent proteome changes by manually dividing the data into sub-group1 (Skin_Ma4w pooled with Skin_Ma14w) and sub-group2 (Skin_Fe4w pooled with Skin_Fe14w) for batch effect correction (in this case, we considered the 2 age groups as different batches). Author response image 1 and D show the comparison with our original data.

**Author response image 1. sa2fig1:** 

Our results show that batch effect correction was able to separate batches based on age or sex differences to a certain extent, e.g. approximately 55% (panels A and B) of DEPs that truly describe age differences based on our real study data, and 92% (panel C) and 45% (Panel D) that truly describe sex differences based on our real study data. However, it should be used with caution, as obtained results (i) miss a significant amount of biological information as shown in Venn diagrams above (white areas in Venn diagrams) and (ii) added different DEPs (dark grey areas in Venn diagrams). Thus, batch correction could only partially replicate our original data generated upon careful age and sex matching in well-controlled experiments. Given that we tested in silico batch correction on prominent age differences, i.e. 4w versus 14w, and observed aforementioned limitations in performance, these limitations will likely impact the performance of correction even more when considering scenarios often used in biomedical studies, e.g. pooling data of mice aged 3-8w or 6-20w.In summary, the here observed limitations of batch correction highlight the importance of adequate experimental design and careful age- and sex-matching in biomedical studies.

2. The authors failed to provide biochemical validation of DEPs identified from the work. I would appreciate that functional testing of the relevant changes to the mouse phenotypes, but this might be beyond the scope of this paper but do wonder whether relying on changes in protein expression level are truly enough to conclude the impact in the preclinical settings.

Indeed, biochemical and functional validation of DEPs is beyond the scope of this manuscript and we are grateful to the reviewers for their understanding.

We agree with the reviewer that a direct/linear relation between changes of the expression of a molecule or group of molecules (at any -omic level) and the phenotype cannot be expected. Yet, the breadth of our datasets offers a different perspective on distinct functional pathways that may be linked to sex and age and, in general, interrogated (patho)physiological changes.

Also, evidence is accumulating that proteome dynamics we have uncovered in our previous studies does indeed have biological significance. For example, in Rouwette et al. we discovered protein up- and downregulation upon chronic pain in mice, amongst others of Protein Disulfide Isomerase (PDI) and proteins belonging to mitochondrial pathways (Rouwette et al. 2016, https://doi.org/10.1074/mcp.M116.058966). To validate the functional relevance of these findings, we injected mice with (i) Rotenone to interfere with the respiratory electron chain complex I and (ii) PACMA31 to inhibit PDIs. Importantly, both treatments resulted in significant changes in pain-associated behaviors compared to controls (Rouwette et al. 2016, https://doi.org/10.1074/mcp.M116.058966). Even more valuable, another independent research group has recently validated the regulation and significance of PDI in pathological pain with complementary methods and studied underlying mechanisms in detail (Zhang et al. 2022, https://doi.org/10.1016/j.celrep.2022.110625).

Furthermore, we have recently published a follow-up paper on one of the proteins found to be regulated upon neuropathic pain in our proteome study (Rouwette et al., 2016, https://doi.org/10.1074/mcp.M116.058966), i.e. the previously uncharacterized transmembrane protein Tmem160. Our functional investigations revealed the significance of Tmem160 for the establishment of nerve injury-induced neuropathic pain in mice (Segelcke et al. 2021, https://doi.org/10.1016/j.celrep.2021.110152)

3. The impact of sex seems to be not impressive based on the PCA analysis of proteome, especially compared to age (Figure 2G and H) and PCA plots using sex-dependent DEPs are misleading (Figure 5C and D).

The reviewer correctly pointed out that sex does not appear as a major contributor to proteome changes in skin and SCN. In particular Figure 2G and H highlight the difference between sex- and age-dependent changes showing the greater impact of age as a biological variable. In addition, in Figure 5C and D we performed PCA using only sex-dependent DEPs to test whether those DEPs are able to segregate sex as a biological variable in all samples. In doing so we followed a commonly used approach to discriminate differences between conditions using DEPs in quantitative proteomic studies (https://doi.org/10.1186/s1201-022-09361-1, https://doi.org/10.1186/s12014-019-9241-5). However, to avoid further confusion and clearly indicate to the reader the minor impact of sex versus age, we have put particular emphasis on explaining this fact in the revised figure legend of Figure 5C and D.

New figure legend 5C and D:

“C-D: Principal component analysis (PCA) using DEPs of sex-dependent comparisons (in contrast to PCA on all identified proteins illustrated in Figure 2G-H) reveals sex as an effective discriminator in paw skin and SCN tissues; females (magenta) and males (cyan).”

4. Although the authors choose paw skin and associated SCN to interpret nociception and pain, in the skin research fields, paw skins are not usually investigated compared to ear and back skins, as there is a highly limited type of dermatoses involving palms and soles in human. In this regard, the current investigation of paw skins may result in the intrinsic limitation for interpretation of the results in translating human dermatoses including psoriasis, atopic dermatitis, and rosacea, which usually spare palms and soles. It would be more interesting and translationally reasonable if the authors select available gene signatures for other dermatoses typically involving palms and soles, such as palmoplantar pustulosis or vesicular hand eczema.

We thank the reviewers for this very helpful suggestion. We have now included new comparisons of our paw skin datasets with those studies that reported molecular signatures of skin diseases that affect the hands or feet. Given the limited public availability of such kind of datasets, we focused on hand-foot psoriasis, palmoplantar pustulosis and vesicular hand eczema. These comparisons can be found in the revised Figure 6-source data 3. We added the following paragraph to the manuscript:

“It is noteworthy that in our study we investigated hairless glabrous skin in mice. However, transcriptomic profiles of human skin diseases used for comparison are mostly derived from human hairy skin – a difference, which needs to be taken into account when interpreting here presented comparisons between mouse and human skin. This is why we additionally selected transcriptomic studies on skin diseases affecting human glabrous skin, i.e. palms and soles, such as hand-foot psoriasis (Ahn et al., 2018), palmoplantar pustulosis (McCluskey et al., 2022) and vesicular hand eczema (Voorberg et al., 2021). Among top gene signatures presented in glabrous skin datasets (2498 DEGs), 928 proteins were quantified in our paw skin datasets and 96 of them showed age and/or sex dependency including several collagens (Figure 6-source data 3; examples are marked with “#” in Figure 6C).”

5. Although the results for age-dependent changes of proteome profiles in paw skins are somewhat predictable (4w: immature skin, 14w: mature skin), sex-dependent difference in paw skin proteome is quite interesting. As female sex is more prone to autoimmune skin diseases such as lupus erythematosus, it would be more interesting if the authors further analyze whether sex-dependent proteome difference is enriched in autoimmune-associated gene signature.

We thank the reviewers for suggesting this highly interesting aspect, which we have followed up in the revised version of the manuscript. We added the following analysis/paragraph to the manuscript:

“Among the skin diseases used here for comparison (Figure 6-source data 1) several are known to exhibit an autoimmune component such as alopecia areata, lichen plantus, lupus erythematosus, psoriasis, and vitiligo. Overall, autoimmune skin diseases are more prevalent in females. Thus, we specifically checked whether top gene signatures of aforementioned autoimmune-associated skin diseases are among the here reported female-enriched proteome changes of our study (Figure 1-source data 1). Indeed, 11 proteins could be identified, of which 9 were differentially regulated by age only in females (Figure 6-source data 1, data sheet 3). For instance, Transforming growth factor-β-induced protein ig-h3 (Tgfbi) and Transcription factor Sp1 (Sp1) were more abundant in 14 weeks female skin, but no significance was found in males.”